# Direct Potential Modulation of Neurogenic Differentiation Markers by Granulocyte-Colony Stimulating Factor (G-CSF) in the Rodent Brain

**DOI:** 10.3390/pharmaceutics14091858

**Published:** 2022-09-02

**Authors:** Judith Kozole, Rosmarie Heydn, Eva Wirkert, Sabrina Küspert, Ludwig Aigner, Tim-Henrik Bruun, Ulrich Bogdahn, Sebastian Peters, Siw Johannesen

**Affiliations:** 1Department of Neurology, University Hospital Regensburg, 93053 Regensburg, Germany; 2Department of Anesthesiology, University Hospital Regensburg, 93053 Regensburg, Germany; 3Institute of Molecular Regenerative Medicine, Paracelsus Medical University Salzburg, 5020 Salzburg, Austria; 4Spinal Cord Injury and Tissue Regeneration Center Salzburg (SCI-TReCS), Paracelsus Medical University, 5020 Salzburg, Austria; 5Velvio GmbH, 93053 Regensburg, Germany; 6Department of Neurology, BG Trauma Center, 82418 Murnau (Staffelsee), Germany

**Keywords:** brain repair, neurodegeneration, neurogenesis, neuronal stem cells, G-CSF, high-dose G-CSF, cellular brain therapy, cerebellum, filgrastim safety

## Abstract

The hematopoietic granulocyte-colony stimulating growth factor (G-CSF, filgrastim) is an approved drug in hematology and oncology. Filgrastim’s potential in neurodegenerative disorders is gaining increasingly more attention, as preclinical and early clinical studies suggest it could be a promising treatment option. G-CSF has had a tremendous record as a safe drug for more than three decades; however, its effects upon the central nervous system (CNS) are still not fully understood. In contrast to conceptual long-term clinical application with lower dosing, our present pilot study intends to give a first insight into the molecular effects of a single subcutaneous (s.c.) high-dose G-CSF application upon different regions of the rodent brain. We analyzed mRNA—and in some instances—protein data of neurogenic and non-neurogenic differentiation markers in different regions of rat brains five days after G-CSF (1.3 mg/kg) or physiological saline. We found a continuous downregulation of several markers in most brain regions. Remarkably, cerebellum and hypothalamus showed an upregulation of different markers. In conclusion, our study reveals minor suppressive or stimulatory effects of a single exceptional high G-CSF dose upon neurogenic and non-neurogenic differentiation markers in relevant brain regions, excluding unregulated responses or unexpected patterns of marker expression.

## 1. Introduction

Human granulocyte-colony stimulating factor (G-CSF, filgrastim)—a member of the hematopoietic growth factor family—was initially characterized as a sole stimulator of hematopoiesis [1]. Filgrastim was originally approved for clinical use in neutropenia and for stem cell mobilization in the early 1990s [2]. Additional evidence is accumulating for G-CSF as a relevant growth factor for neurogenesis in embryonal development [3]. Thus, G-CSF is seen as a promising treatment candidate in degenerative or traumatic brain disorders, which still have only limited treatment options [4].

G-CSF has already been successfully applied in animal models, where an increase in neurogenesis could be demonstrated [5,6,7]. Similarly, preliminary clinical studies have shown promising results in humans, including in the treatment of amyotrophic lateral sclerosis (ALS) [8,9,10]. By mobilizing hematopoietic stem cells through G-CSF treatment, a very elegant technique is gaining more and more attraction in stem cell associated treatment schemes for neurodegenerative or post-traumatic disorders of the brain and spinal cord [11,12]. Specifically, the results in ALS patients have elicited new perspectives for a successful treatment avenue [13]. In all these clinical translations, dosing of G-CSF was applied within the range of hitherto established and safe hematologic indications.

Neurogenesis, the generation of functionally active new neurons, is an essential requirement in the developing brain and spinal cord, and relevant active neurogenesis remains during adulthood, even in humans [14,15], with ongoing vibrant controversies concerning affected age groups, brain regions, extent, and clinical relevance [16,17]. The effects of G-CSF on the different stages of neuronal differentiation, including different neurogenesis markers, as well as effects on glial differentiation and proliferation markers in different relevant brain regions have, to the best of our knowledge, not yet been fully examined. This would help to understand the clinically observed heterogenous response and in case of higher dosing, also the safety in prospective central nervous system (CNS) applications.

We therefore initiated a small pilot study to elucidate these mechanisms further. We selected different brain regions for analysis of mRNA and proteins to gain a first insight into the influence of G-CSF on the rodent CNS. In essence, we administered a single high dose of G-CSF to a small number of rats and subsequently analyzed the brains after five days. We selected brain areas where adult neurogenesis has already been demonstrated, such as the subventricular zone and the hippocampus [18,19], as well as other areas, such as the cerebellum, corpus callosum, and hypothalamus. For a more comprehensive overview samples, were also isolated from other brain areas, associated with movement, such as motor cortex, substantia nigra, pyramidal tracts, putamen, and also nucleus accumbens and the prefrontal cortex. We focused on markers for different stages of neurogenesis (Nestin (Nes) [20], Neuronal Nuclei (NeuN) [21], SRY (sex determining region Y)-box 2 (SOX2) [22], Musashi-1 (MSI1) [23], Glial fibrillary acidic protein (GFAP) [24], Doublecortin (DCX) [25], Neural cell adhesion molecule (NCAM) [26], and Class III β-tubulin (TUBB3) [27]), as well as the general proliferation marker Kiel Antigen Nr. 67 (Ki 67) [28] and markers for hematopoietic stem cells (CD34 [29]), transforming growth factor (TGFB) signaling [30], oligodendroglial cells (oligodendrocyte transcription factor (Olig2) [31]) microglial cells (allograft inflammatory factor 1 (AIF-1) = ionized calcium-binding adapter molecule 1 (IBA-1) [32]), and G-CSF receptor CSF3R [33].

## 2. Materials and Methods

### 2.1. Animals

Male Fisher-344 rats (Rattus norvegicus) (Charles River, Sulzfeld, Germany) weighing 240–300 g were housed in groups of three in standard polycarbonate rat cages (40 × 60 × 20 cm) under standard laboratory conditions (12-h light/dark cycle, lights on at 06:00 AM, 22 °C, 60% humidity) with ad libitum access to food and water. All experimental protocols were approved by the Committee on Animal Health and Care of the government Lower Franconia, Germany, and conformed to international guidelines on the ethical use of animals (protocol code 55.2 DMS-2532-2-245, date of approval: 27 June 2016). All efforts were made to minimize the number of animals used and their suffering.

### 2.2. Experimental Procedure

Schematic illustration of the experimental procedure is shown in Figure 1. Following injection of G-CSF (Filgrastim: 1.3 mg/kg; s.c., Hexal, Holzkirchen, Germany) or vehicle (NaCl: 1 mL/kg; s.c., Braun, Braunschweig Germany) rats underwent repeated blood samplings via the lateral tail vein according to the experimental procedure scheme. Each group consisted of six animals. In order to investigate whether a single s.c. G-CSF injection influences the neurogenic niche activity, differentiation patterns of neuronal cells, hematopoietic stem cells, oligodendroglial cells, microglial cells, and the central TGFB system, brains were rapidly removed on day 5 after drug administration, snap-frozen in methylbutane, cooled on dry-ice, and stored at −80 °C for subsequent mRNA and protein expression analysis.

### 2.3. Cutting of Brains

After embedding brains in Tissue-Tek (Sakura, Torrance, CA, USA), they were cut into 500 µm slices using a cryostat with a temperature of −20 °C. From these slices, regions were isolated with a scalpel and immediately put on ice in an autoclaved 1.5 mL cup. Material from the left hemisphere was used for isolation of proteins, and from the right hemisphere for isolation of mRNA. Anatomical regions were clearly identified by using a brain atlas [34]. The scalpel was cleaned with 99.8% ethanol between steps. Samples were stored at −20 °C until further use.

### 2.4. Isolation of RNA

RNA was isolated with innuPREP RNA Mini Kit (Analytik Jena, Jena, Germany), according to protocol. The amount of RNA in the samples was measured with a photometer (Eppendorf Biophotometer D30, Eppendorf SE, Hamburg, Germany). Then, samples were stored at −80 °C until further use.

### 2.5. DNase Digest

To prevent gDNA contamination, every sample was treated with innuPREP DNase I Kit (Analytik Jena) according to protocol.

### 2.6. Isolation of Proteins

For smaller amounts of tissue, 200 µL of T-PER^®^ Tissue Protein Extraction Reagent (Thermo Scientific, Braunschweig, Germany) was added. For larger amounts, up to 500 µL of T-PER^®^ was used. Then the samples were pestled until the tissue was in solution, followed up from centrifugation at 12,000 rpm for five minutes. Samples were stored at −80 °C until further use.

### 2.7. Quantitative Real-Time Polymerase Chain Reaction (qRT-PCR)

For mRNA analysis, following determination of RNA content (100 ng RNA per 20 μL), the RNA was reversely transcribed into first strand cDNA with iScript cDNA Synthesis Kit (BioRad, Hercules, CA, USA) according to manufacturer’s recommendations. For mRNA analysis, qRT-PCR was performed using a CFX96 Touch Real Time PCR Detection System (BioRad, Hercules, CA, USA). All primer pairs (CD34 (qRnoCED0055141), DCX (qRnoCID0004149), GFAP (qRnoCED0005713), MKi67 (qRnoCID0001488), MSI1 (qRnoCID0009554), Nes (qRnoCED0004560), TGFB1 (qRnoCID0009191), TGFB2 (qRnoCID0006448), TGFBR2 (qRnoCID0006991), NCAM1 (qRnoCID0009053), Aif1 (qRnoCED0002514), CSF3R (qRnoCED0007571), Rbfox3 (qRnoCED0019388), Olig2 (qRnoCED0005985), SOX2 (qRnoCED0005008), Tubb3 (qRnoCED0051163), and GAPDH (qRnoCID0057018)] were ready-to-use standardized and were mixed with the respective ready-to-use Mastermix solution (Sso Advanced Universial SYBR Green Supermix, BioRad, Hercules, CA, USA) according to manufacturer’s instructions (BioRad Prime PCR Quick Guide). As template, 0.2 μL of respective cDNA was used. H2O was used as a negative control for qRT-PCR. For relative quantification, housekeeping gene GAPPDH was used. Afterward, BioRad CFX Manager 3.1 was used to quantify mRNA-level relative to GAPDH mRNA [35].

### 2.8. Western Blot

Protein concentrations were determined using Pierce Coomassie Plus Assay Kit (Life Technologies, Carlsbad, CA, USA). SDS-acrylamid-gels (10%/12%) were produced using TGX Stain Free™ Fast Cast™ Acrylamid Kit (BioRad, Hercules, CA, USA) according to manufactory instructions. Protein samples (20 μL) were diluted 1:4 with Lämmli-buffer (6.5 μL, Roti^®^-Load1, Roth, Karlsruhe, Germany), incubated at 60 °C for 30 min and loaded on the gel with the entire volume of the protein solution. Separation of proteins was performed by electrophoresis using Power Pac Basic Power Supply (BioRad, Hercules, CA, USA) and Mini Protean Tetra cell electrophoresis chamber (BioRad, Hercules, CA, USA) (200 V, 45 min). Following electrophoresis, the proteins were blotted using Trans-Blot Turbo Transfer System (BioRad, Hercules, CA, USA). All materials for Western blotting were included in Trans Blot Turbo RTA PVDF-Midi Kit (BioRad, Hercules, CA, USA). The PVDF-membrane for blotting procedures was activated in methanol (Merck, Darmstadt, Germany) and equilibrated in 1× transfer buffer. Following blotting (25 V, 1 A, 30 min), membranes were washed (3 × 10 min, room temperature (RT)) with 1× TBS (Roth, Karlsruhe, Germany) containing 0.05% Tween-20 (Roth, Karlsruhe, Germany). Afterward, the membranes were blocked with 5% BSA (Albumin-IgG-free, Roth, Karlsruhe, Germany), diluted with TBS-T for 1 h at RT, the primary antibodies (diluted in 0.5% BSA in TBS-T) were added and incubated at 4 °C for 2 days [rabbit anti-Nes (1:1000, Abcam), rabbit anti-SOX2 (1:2000, Abcam), rabbit anti-MSI1 (1:2000, Abcam), rabbit anti-GFAP (1:5000, Abcam), rabbit anti-DCX (1:1000, Cell Signaling), mouse anti-NCAM (1:2000, Cohesion), rabbit anti-NeuN (1:2000, Abcam), mouse anti-TUBB3 (1:3000, Covalab), rabbit anti-CD34 (1:1000, Abcam), rabbit anti-TGFB1 (1:500, LSBio), mouse anti-TGFBR2 (1:500, Santa Cruz)]. Next, membranes were washed in TBS-T (3 × 10 min, RT) and incubated with the secondary antibody (anti-rabbit HRP, or anti-mouse HRP, Cell Signaling) 1:10,000 + Strep-Tactin HRP, 1:12,500) (1 h, RT). Following incubation, blots were washed with TBS-T, emerged using Luminata™Forte Western HRP Substrate (Millipore, Millipore, Germany) and bands were detected with a luminescent image analyzer (ImageQuant LAS 4000, GE Healthcare, Munich, Germany). Afterward, the blots were washed in TBS-T (3 × 10 min, RT) and blocked with 5% MP diluted in TBS-T (1 h, RT). For housekeeper comparison, the membranes were incubated with HRP-conjugated anti GAPDH (1:2000 in 0.5% BSA, 4 °C, overnight; Cell Signaling). On the following day, blots were emerged using Luminata™Forte Western HRP Substrate (Millipore, Darmstadt, Germany) and bands were detected with a luminescent image analyzer (ImageQuant LAS 4000, GE Healthcare, Munich, Germany). Finally, the blots were washed with TBS-T (3 × 5 min) and stained using 1× Roti Blue solution (Roth, Karlsruhe, Germany) and dried at RT. Blots were analyzed using Image Studio Lite Software Version 5.2 (Licor, Lincoln, NE, USA) [35].

### 2.9. Statistics

GraphPad Prism 8 was employed for graph design and statistical comparisons. All parameters were tested for Gaussian distribution using D’Agostino–Pearson omnibus normality test. Then, all parameters were analyzed using a two-tailed Student’s *t*-test or the Mann Whitney test, depending on Gaussian distribution. Data are presented as median with min to max. Significance was observed at *p* ≤ 0.05 (*), *p* ≤ 0.01 (**), *p* ≤ 0.001 (***), and a trend was noticed at *p* ≤ 0.1 [35].

## 3. Results

To investigate the effects of a single dose G-CSF on the rodent central nervous system, we isolated mRNA of the brain regions listed in Figure 2 and analyzed it by qRT-PCR. For a comprehensive functional overview, qRT-PCR data were supplemented by a protein analysis of a smaller selection of relevant markers in tissue samples from the subventricular zone, corpus callosum, and cerebellum via Western blot. The Figure 3, Figure 4, Figure 5, Figure 6 and Figure 7 provide an overview of the significant changes in regions of particular interest. A complete overview of results in all regions is shown in the Appendix A. As a functional reference, leukocyte counts were recorded in G-CSF and NaCl treated rats at several time points after the single injection. As expected, G-CSF treated animals displayed a clear leukocytosis as a mobilization effect.

### 3.1. Differentiation Markers in Classic Anatomical Areas of Neurogenesis

#### 3.1.1. Subventricular Zone

The mRNA expression of Nestin in the subventricular zone did not differ significantly between the control group (NaCl 0.9%) and the group treated with G-CSF. As shown in Figure 3, the other markers for early-stage neurogenesis, SOX2 (*p* = 0.0233) and MSI1 (*p* = 0.0101), were downregulated in the G-CSF treated rats. Additionally, the expression of NCAM (*p* = 0.0131) and TUBB3 (*p* = 0.039) was decreased in the G-CSF group. A trend towards downregulation was observed for Ki67 (*p* = 0.0778) as well as NeuN (*p* = 0.0575). The expression of oligodendrocyte marker Olig2 (*p* = 0.0115) and microglial marker AIF-1 (*p* = 0.0056) was significantly lower compared to the control group. Moreover, the CSF3R showed a significant downregulation (*p* = 0.0173). No significant changes were observed within the TGFB signaling system. Due to a limited amount of biopsy material from the subventricular zone western blot analysis was limited to the markers SOX2, MSI1, GFAP; DCX, NCAM, NeuN, TUBB3, and CD34, where no changes were observed except for a trend towards a downregulation of TUBB3 (*p* = 0.0906).

#### 3.1.2. Hippocampus

Within the hippocampus, as shown in Figure 4, the only significant changes observed were an upregulation of Ki67 (*p* = 0.0317) and a downregulation of TUBB3 (*p* = 0.0260). The markers of early neurogenesis did not significantly differ between the groups. Trends towards a downregulation of Olig2 (*p* = 0.0584), TGFB2 (*p* = 0.0926), and CSF3R (*p* = 0.0736) were observed.

### 3.2. Differentiation Markers in Further Anatomic Regions

#### 3.2.1. Corpus Callosum

Within the corpus callosum the mRNA of stem cell marker Nestin showed a trend towards a downregulation in the G-CSF treated group (*p* = 0.0563), whereas SOX2 (*p* = 0.0239) and MSI1 (*p* = 0.0152) were significantly downregulated, as shown in Figure 5. The expression of GFAP did not differ between groups. While immature neurons’ marker DCX showed significantly downregulated mRNA levels (*p* = 0.0131), the level of TUBB3 displayed a trend towards downregulation (*p* = 0.0618). Proliferation marker Ki67 was noticeably less expressed within the G-CSF treated group (*p* = 0.0180). The marker for mature neurons, NeuN, showed a significant downregulation within the treatment group (*p* = 0.0269). Hematopoietic stem cell marker CD34 did not differ significantly but showed a trend towards a lower expression within the G-CSF group (*p* = 0.0622). TGFB1 was significantly downregulated (*p* = 0.0190), while we detected a trend in downregulation of TGFB2 (*p* = 0.0529) and TGFBR2 (*p* = 0.0525). The CSF3R was not affected. Data of significantly downregulated oligoglial marker Olig2 (*p* = 0.0111) as well as microglial marker AIF-1 (*p* = 0.0303) are shown in the Appendix A.

Protein analysis supported the already observed downregulation of mRNA with a significant downregulation of MSI1 (*p* = 0.0377). Nestin, SOX2, and GFAP did not differ from control group. DCX protein levels showed a trend towards downregulation (*p* = 0.0886). Like mRNA, protein levels of TUBB3 indicated a negative trend (*p* = 0.0931). NeuN did not show a significant change. CD34 did not differ. TGFB1was significantly downregulated (*p* = 0.0281), although this might be due to an outlier in the control group. TGFBR2 did not show any significant difference.

#### 3.2.2. Cerebellum

The cerebellum, results shown in Figure 6, holds an opposing position in our study by its numerous upregulation in contrast to the almost continuous downregulation of various markers observed in the other regions. Nestin showed a significantly higher expression within the treatment group (*p* = 0.0281). Also, SOX2 (*p* = 0.0241) and MSI1 (*p* = 0.0384) were noticeably upregulated, equally as GFAP (*p* = 0.0299). DCX and NCAM expression did not differ between groups, whereas TUBB3 was significantly upregulated (*p* = 0.0476). Ki67 could not be analyzed properly due to lack of material in sufficient quality. Olig2 (*p* = 0.0866) (data shown in the Appendix A) and CD34 (*p* = 0.0724) both showed a trend towards upregulation. The expression of AIF-1 was not affected. TGFB1 (*p* = 0.0180) and TGFBR2 (*p* = 0.0260) were significantly upregulated, while upregulation of TGFB2 was indicated by a trend (*p* = 0.0649). CSFR did not show a difference between the two groups.

The protein level of Nestin was downregulated (*p* = 0.0007), discordant with the mRNA level. Furthermore, a significant upregulation of NeuN protein, concordant with mRNA data, was observed (*p* = 0.0346). The other western blot data did not show significant changes.

#### 3.2.3. Hypothalamus

As in the cerebellum, we also detected a trend for upregulation of Nestin (*p* = 0.0844), NeuN (*p* = 0.0837), TUBB3 (0.0813), and Olig2 (*p* = 0.0886) in the hypothalamus, shown in Figure 7. The other markers were not altered, with exception of CD34, which showed a significantly higher expression in the treatment group (*p* = 0.0076).

#### 3.2.4. Prefrontal Cortex

Due to poor quality, we could only analyze a small number of samples from the prefrontal cortex. NCAM was significantly downregulated (*p* = 0.0229), although we have to point out the small number of only three animals in the control group and four animals in the treatment group. While TGFB1 showed a trend towards a downregulation (*p* = 0.0729), TGFB2 (*p* = 0.0409), and TGFBR2 (*p* = 0.0232) had a significantly lower expression in the treatment group. Graphs are shown in the Appendix A.

#### 3.2.5. Motor Cortex

The only significant change obtained from samples of the motor cortex was a lower expression of AIF-1 within the G-CSF treated group (*p* = 0.0022).

#### 3.2.6. Putamen

Samples from the putamen of G-CSF treated animals had significantly lower mRNA levels of NCAM (*p* = 0.0004) as well as CD34 (*p* = 0.0086) compared to the control group. MSI1 showed a trend towards downregulation (*p* = 0.0801).

#### 3.2.7. Nucleus Accumbens

SOX2 was significantly downregulated in the nucleus accumbens within the treatment group (*p* = 0.0402). Furthermore, expressions of TUBB3 (*p* = 0.0212) and NeuN (*p* = 0.0002) were significantly lower. Additionally, the oligodendrocyte marker Olig2 showed a noticeable significant downregulation (*p* = 0.0040). TGFBR2 showed a trend towards downregulation (*p* = 0.0726).

#### 3.2.8. Substantia Nigra

Significant downregulation was seen for DCX (*p* = 0.0388) and TGBR2 (*p* = 0.0137) in substantia nigra of G-CSF treated animals. A trend towards downregulation of GFAP (*p* = 0.0839) and CD 34 (*p* = 0.0703) was observed. However, the small number of animals must be mentioned as a limitation.

#### 3.2.9. Pyramidal Tracts

While Nestin and MSI1 did not differ between groups in samples of the pyramidal tracts, SOX2 showed a trend towards downregulation in the treatment group (*p* = 0.0802). GFAP (*p* = 0.0152) and DCX (*p* = 0.0257) showed a significantly lower expression. While NCAM indicated a trend towards downregulation (*p* = 0.0722), TUBB3 was significantly downregulated (*p* = 0.0476). The mature neurons’ marker NeuN also indicated a trend towards downregulation (*p* = 0.0847). Olig2 showed a significantly lower expression within the treatment group (*p* = 0.0390), the same applied to CSF3R (*p* = 0.0130).

## 4. Discussion

The intention of this early pilot trial was to elucidate a potential effect of a single high dose of the hematopoietic growth factor G-CSF [1] upon neurogenic and non-neurogenic differentiation markers of the rodent brain. Effects on some oligodendrocyte, microglial, and hematopoietic markers were also analyzed. Several anatomical regions were selected to detect the potential modulation of local differentiation activities. The results of the present study suggest that a single high dose of G-CSF neither leads to an unregulated response of differentiation markers in all observed brain regions nor to an unexpected new pattern of expression within these areas of the rodent brain.

Surprisingly the effects were regulated differently in the various neurogenic regions. We identified few regions with almost no regulation. The majority of regions had either up- or downregulated marker signals, with the direction of regulation being consistent for neuronal and non-neuronal differentiation markers (see Figure 2). The observed effects were mainly substantiated by mRNA data—in several cases they were confirmed by protein analysis. However, frequently, protein data were either not available or non-significant. When significantly altered, they were partly in line with mRNA data. Therefore, we focused our analysis mainly upon mRNA results.

We observed a downregulation of neuronal differentiation markers over all stages within corpus callosum, putamen, subventricular zone, nucleus accumbens, and pyramidal tracts. This was mapped by several markers. The same applied for each single tested oligodendroglial, microglial, and interestingly, also hematopoietic stem cell marker. The cerebellum and hypothalamus form a remarkable contrast to these other regions studied. Especially, the cerebellum showed a noticeable significant upregulation of several markers. Most results from the hypothalamus were less apparent. Aside from the significant upregulation of CD34, other markers obtained here were upregulated by a trend, thus resembling the cerebellar findings. These noticeable effects lead to several questions.

Special attention should be given to the already mentioned nearly continuous up- or downregulation of different stages of neuronal differentiation, of oligodendrocytes, microglia, and hematopoietic stem cells in the regions tested. The homogeneity of the response to G-CSF in the different brain regions most likely reflects a specific cell response to this particular growth factor. Cell response to a growth factor and markers expressed during neurogenesis change dynamically over time and are physiologically dependent on a highly regulated sequential differentiation process [40]. As we only obtained samples from a single timepoint with a single high-dose treatment, it was interesting to see (1) patterns of region specific down- or upregulation, (2) a homogenous up- or downregulation across the marker spectrum, and (3) an inline behavior of hematopoietic markers with the neurogenic and non-neurogenic marker spectrum. However, we did not observe an unexpected pattern of neurogenic or non-neurogenic markers within their respective differentiation lines. Even with the relatively low number of experimental animals, these observations seem to be consistent. With the delay of five days after single treatment, it remains a future task to determine the long-time effects of G-CSF on neuronal markers, respectively neurogenesis. Concerning the observed wide spread downregulation, it may be possible that day five already reflects a negative cellular feedback response or counteraction and that sample acquisition at an earlier timepoint would reveal an upregulation. This can only be assessed in a larger scale study with a higher number of animals, different dosing, and extended times of sampling.

However, why do the findings within the cerebellum and hypothalamus differ so substantially from the findings in the other regions? Due to the standard laboratory conditions the rats are living in, they may only be challenged by limited space and stressed by companion animals. Therefore, both regions—cerebellum and hypothalamus—may be involved in high functional activity. Under these circumstances, it is reasonable that G-CSF might hit functionally active and demanding regions there. This may be completely different in an enriched environment, which should be planned to be involved in further experiments. Without G-CSF stimulation, the enriched environment has already been described to lead to a generation of oligodendrocytes in the sensorimotor cortex and corpus callosum [41] and neurons in the cerebellum [42], as well as in the hypothalamus [43], among numerous other findings. In both outstanding regions of upregulation in this study, the cerebellum and hypothalamus, there is previous evidence of neuronal differentiation [44]. Within the cerebellum, the neuronal process of adult neurogenesis remains uncertain, though neuronal proliferation has been described, especially in the injured murine brain [45] and after physical exercise [42]. Neurogenesis has also been reported to occur in the hypothalamus [46], among others, to regulate energy balance [47].

Surprisingly, although hippocampal neurogenesis is broadly accepted [48,49], the results within the hippocampus show little change at all, as shown in Figure 4. A significant upregulation of Ki67 is an exception with remarkable outliers. Regrettably, we did not perform protein analysis here to supplement the mRNA data.

Aside from the hippocampus, the subventricular zone is also considered a classical neurogenic site [50,51]. Neurogenesis can be stimulated here through many different factors, including intrinsic and extrinsic, neuronal disorders, and specifically motor exercise [44]. We again observed a downregulation of neuronal differentiation markers in the subventricular zone, including oligodendroglial, microglial, and hematopoietic markers with this single high-dose treatment.

The chosen conditions of the performed pilot study at a single time point of analysis are clearly a major limitation for interpretation and conclusion. Neuronal and non-neuronal differentiation is a complex process over time requiring a sequence of subsequent events—demanding similar time points of analysis. The surprising downregulation of several markers of neurogenesis in traditional neurogenic niches underlines the future need to focus on the temporal development in up- and downregulation of the given markers over time. In addition to mRNA data, a systematic protein and single cell transcriptome analysis in parallel would be supportive. A larger number of experimental animals, additional stimuli such as an enriched environment group, and drug administration over a longer period of time as well as different dosing could assist in clarifying the functional role of G-CSF in neurogenesis. However, the intention of the present pilot study, apart from giving a first insight into the effects of G-CSF upon neuronal markers, was to exclude an unexpected neuronal precursor reaction after an accidental or unintended G-CSF overdose.

Different reviews show preclinical studies usually using doses from 5–300 µg/kg [4,12,52]. An extreme dose of 1665 µg/kg had been used in a mouse model of stroke, altering monocyte trafficking [53]. Our intention was to use a high dose of G-CSF in the range of approximately 100-fold compared to human (5–10 µg/kg) and frequent animal dosing (see above). Considering this as a somewhat arbitrary decision, it reflects our intention for a preliminary risk estimation in this pilot study. In the same context, the time point of five days after treatment was chosen for analysis to see most of the immediate and also ongoing cell responses, minimizing the required number of experimental animals. Generally, from its use in hematology and oncology, G-CSF is known to be a safe drug, with its main adverse effect bone pain [54]. Collectively, these findings reveal that even a very high single G-CSF dose (corresponding to 1.3 mg/kg)—which might be the consequence of an accidental overdose in prospective clinical schemes—does not lead to a hazardous consequence upon neurogenic or non-neurogenic differentiation markers of the CNS. Region specific variations of activation are negligible from a safety perspective, but may indeed reflect the specific situation of the experimental animals concerning brain activation. Of interest is also the inline activation pattern of the examined hematopoietic differentiation marker CD34, delivering a further surplus of safety information. The pattern of activation in the classical inflammatory response of the TGFB-system may be interpreted in the same direction. Taking into account the significant evidence of systemic and organ specific safety for filgrastim [54], our pilot trial of high-dose G-CSF exposure does not reveal any new considerable CNS safety issues. Therefore, these data on neurogenic differentiation markers may contribute additional confidence for G-CSF as a future new treatment perspective in the wide spectrum of neurodegenerative or post-traumatic brain disorders.

## Figures and Tables

**Figure 1 pharmaceutics-14-01858-f001:**
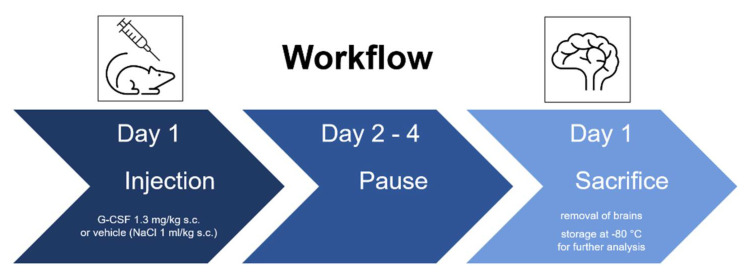
Outline of experimental procedure. Injections (subcutaneous (s.c.)) of granulocyte-colony stimulating factor (G-CSF) (1.3 mg/kg) or vehicle 0.9% NaCl (1 mL/kg) were performed to observe the effects of a single G-CSF administration upon the rodent brain. Rats were sacrificed on day 5 after drug administration, brains were removed, and stored at −80 °C for subsequent mRNA and protein expression analysis.

**Figure 2 pharmaceutics-14-01858-f002:**
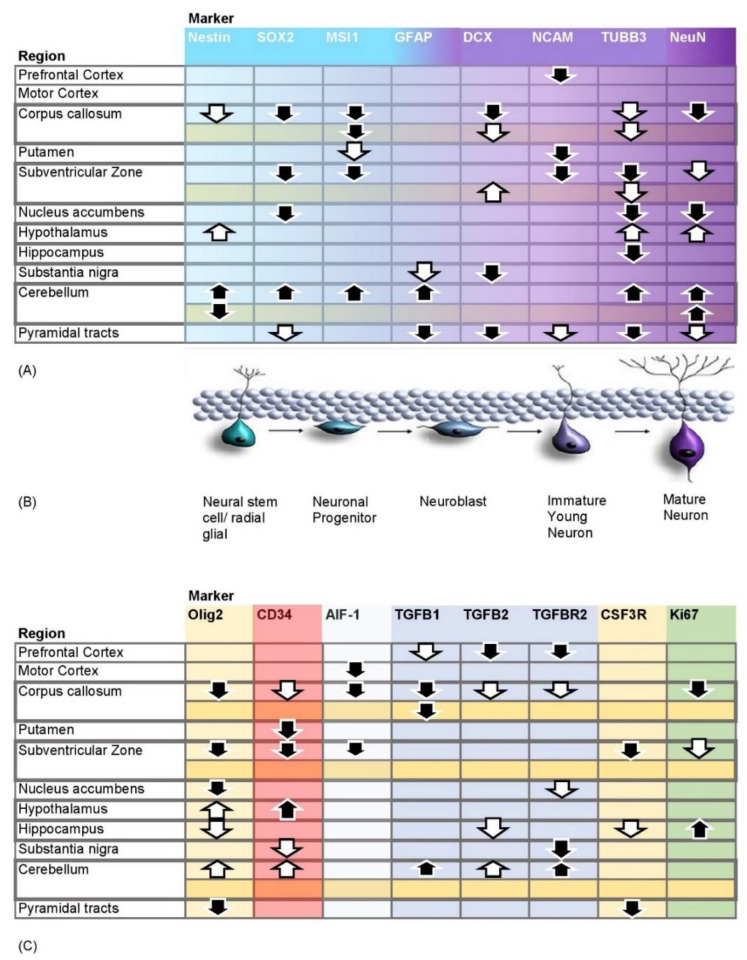
Overview of regions and markers studied. (**A**,**C**) provide an overview of all studied regions and markers. Downward pointing black arrows with white border show significant downregulation (*p* = ≤ 0.05) in the corresponding brain region, upward pointing black arrows with white border show significant upregulation (*p* = ≤ 0.05). Downward pointing white arrows with black border show a trend towards downregulation (*p* = ≤ 0.1), upward pointing black arrows with white border show a trend towards upregulation (*p* = ≤ 0.1). (**A**) shows the markers of neuronal differentiation in chronological order from neural stem cells to mature neurons. Neuronal differentiation is illustrated in (**B**). (**C**) shows the non-neuronal markers. Rows show results of mRNA, except for the second rows of each corpus callosum, subventricular zone and cerebellum, highlighted in yellow, showing the protein results.

**Figure 3 pharmaceutics-14-01858-f003:**
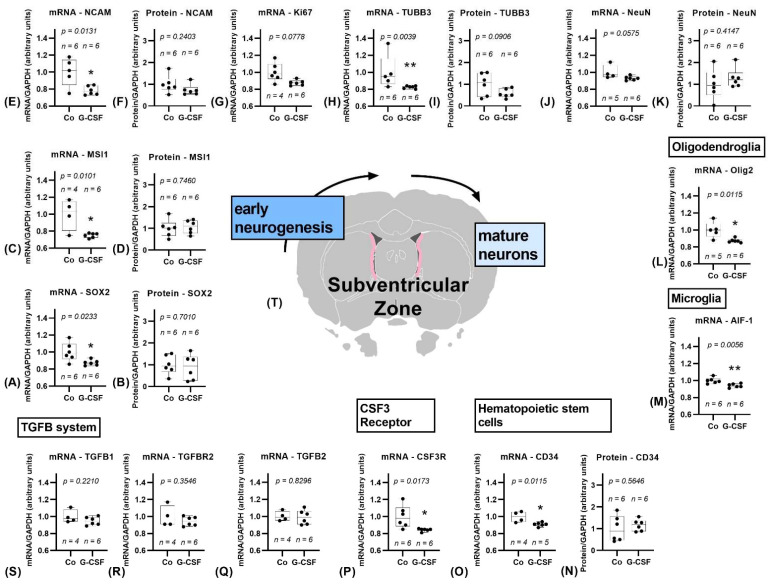
Study results of subventricular zone. The subventricular zone is marked pink in the cross section of the brain (**T**), position 0.43 mm anterior to bregma, image modified from Scalable Brain Atlas [36,37,38,39]. Controls are depicted left, G-CSF-treated animals are displayed on the right side of each graph. Significant downregulation of mRNA for markers of early neurogenic stages (**A**–**E**). mRNA of SOX2 (**A**), MSI1 (**C**), and NCAM (**E**) is significantly downregulated, protein shows no change (**B**,**D**,**F**). Ki67 shows a trend towards downregulation (**G**). mRNA of TUBB3 shows a significant downregulation (**H**), protein a trend towards downregulation (**I**). mRNA of NeuN shows a trend towards downregulation (**J**), protein has not changed (**K**). Oligodendroglial marker Olig2 is significantly downregulated (**L**). Microglial marker AIF-1 is significantly downregulated (**M**). mRNA of hematopoietic stem cell marker CD34 shows a significant downregulation (**O**), protein shows no change (**N**). CSF3 Receptor shows a significant downregulation (**P**). TGFB system is not affected (**Q**–**S**). All parameters were tested for Gaussian distribution using the D’Agostino–Pearson omnibus normality test. Afterward, all parameters were analyzed using a two-tailed Student’s *t*-test or Mann–Whitney test, depending on Gaussian distribution. Data are presented as median with min to max. Significance was observed at *p* ≤ 0.05 (*), *p* ≤ 0.01 (**), a trend was noticed at *p* ≤ 0.1. Numbers (n) are given for each group.

**Figure 4 pharmaceutics-14-01858-f004:**
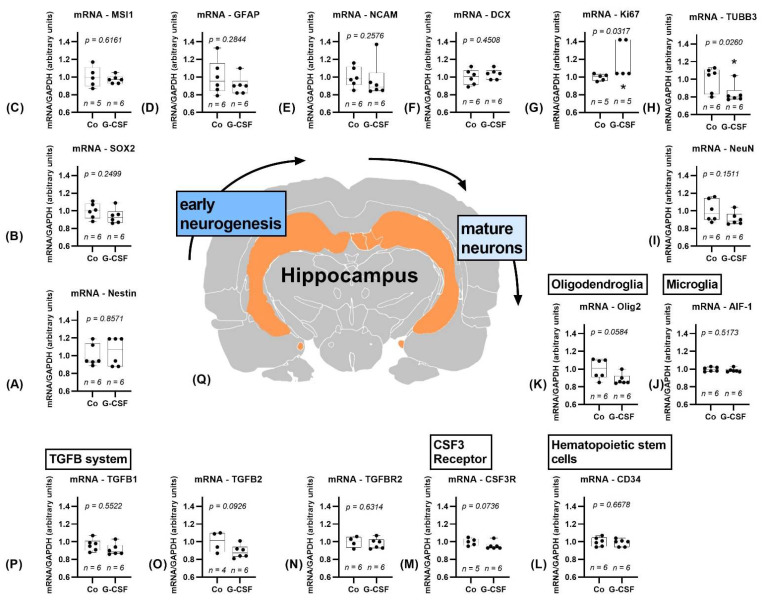
Study results of hippocampus. Hippocampus is marked orange in the cross section of the brain (**Q**), position 3,63 mm posterior to bregma, image modified from Scalable Brain Atlas [36,37,38,39]. Controls are depicted left, G-CSF-treated animals are displayed on the right side of each graph. (**A**–**I**) show results for markers of neuronal differentiation from early neurogenesis to mature neurons. Only Ki67 (**G**) shows a significant upregulation, TUBB3 (**H**) a significant downregulation. Olig2 shows a trend towards downregulation (**K**). AIF-1 (**J**) and CD34 (**L**) do not differ between groups. CSF3 Receptor shows a trend towards downregulation (**M**). TGFB does not show significant changes (**N**–**P**), except for a negative trend of TGFB2 (**O**). All parameters were tested for Gaussian distribution using the D’Agostino–Pearson omnibus normality test. Afterward, all parameters were analyzed using a two-tailed Student’s *t*-test or Mann–Whitney test, depending on Gaussian distribution. Data are presented as median with min to max. Significance was observed at *p* ≤ 0.05 (*), a trend was noticed at *p* ≤ 0.1. Numbers (n) for each group are given.

**Figure 5 pharmaceutics-14-01858-f005:**
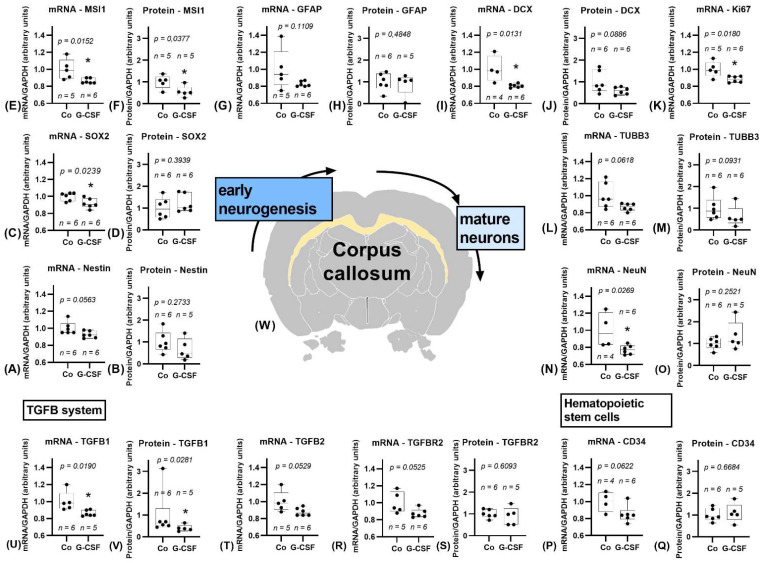
Study results of corpus callosum. Corpus callosum is marked yellow in the cross section of the brain (**W**), position 1.45 mm posterior to bregma, image modified from Scalable Brain Atlas [36,37,38,39]. Controls are depicted left, G-CSF-treated animals are displayed on the right side of each graph. (**A**–**O**) show results for markers of neuronal differentiation from early neurogenesis to mature neurons for mRNA and protein. Nestin shows a trend towards downregulation of mRNA (**A**), protein does not differ (**B**). mRNA data of SOX2 (**C**) and MSI1 (**E**) are significantly downregulated. Protein data of SOX2 (**D**) is not significant, while MSI1 shows a significant downregulation (**F**). GFAP is not significantly changed in mRNA (**G**) and protein (**H**). mRNA of DCX (**I**) is downregulated, protein shows a trend towards downregulation (**J**). mRNA of Ki67 is significantly downregulated (**K**). mRNA (**L**) and protein (**M**) of TUBB3 show a trend towards downregulation. mRNA (**N**) of NeuN is significantly downregulated, protein (**O**) does not differ. Hematopoietic stem cell marker CD34: mRNA shows a trend towards downregulation (**P**), protein does not differ between the groups (**Q**). TGFB1 mRNA (**U**) and protein (**V**) are significantly downregulated, although in (**V**) this might be due to an outlier. mRNA of TGB2 (**T**) shows a trend towards downregulation. TGFBR2 mRNA (**R**) shows a trend towards downregulation, protein (**S**) does not differ. All parameters were tested for Gaussian distribution using D’Agostino–Pearson omnibus normality test. Afterward, all parameters were analyzed using a two-tailed Student’s *t*-test or Mann–Whitney test, depending on Gaussian distribution. Data are presented as median with min to max. Significance was observed at *p* ≤ 0.05 (*), a trend was noticed at *p* ≤ 0.1. Numbers (n) for each group are given.

**Figure 6 pharmaceutics-14-01858-f006:**
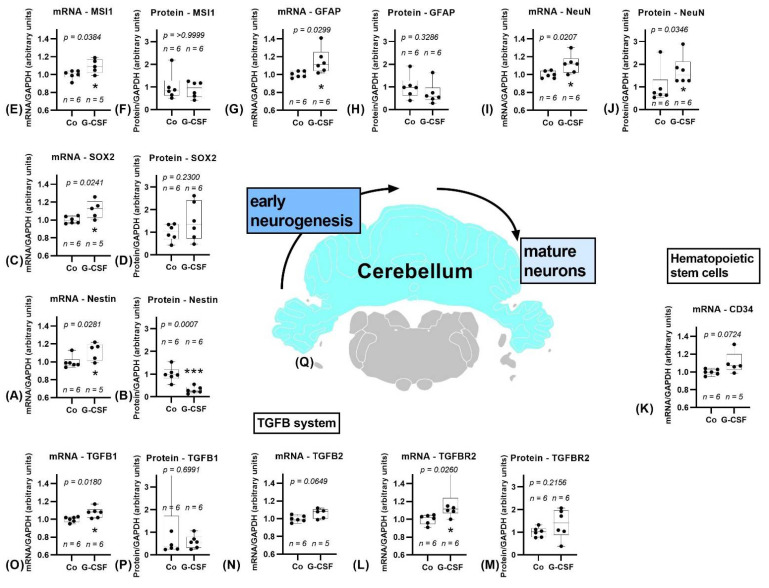
Study results of cerebellum. Cerebellum is marked blue in the cross section of the brain (**Q**), position 11.13 mm posterior to bregma, image modified from Scalable Brain Atlas [36,37,38,39]. Controls are depicted left, G-CSF-treated animals are displayed on the right side of each graph. Significant upregulation of mRNA for markers of early neurogenic stages (**A**–**E**). mRNA of Nestin (**A**), SOX2 (**C**), and MSI1 (**E**) is significantly upregulated, protein data of Nestin is significantly downregulated (**B**), SOX2 (**D**) and MSI1 (**F**) do not differ. GFAP mRNA is significantly upregulated (**G**), protein does not show a difference (**H**). Both mRNA (**I**) and protein (**J**) of NeuN show a significant upregulation. Hematopoietic stem cell marker CD34 shows a trend towards upregulation (**K**). TGFB1 mRNA is significantly upregulated (**O**), protein does not differ (**P**). TGFB2 shows a trend (**N**). mRNA of TGFBR2 is significantly upregulated (**L**), protein does not show a difference (**M**). All parameters were tested for Gaussian distribution using D’Agostino–Pearson omnibus normality test. Afterward, all parameters were analyzed using a two-tailed Student’s *t*-test or Mann–Whitney test, depending on Gaussian distribution. Data are presented as median with min to max. Significance was observed at *p* ≤ 0.05 (*), and *p* ≤ 0.001 (***), a trend was noticed at *p* ≤ 0.1. Numbers (n) are given for each group.

**Figure 7 pharmaceutics-14-01858-f007:**
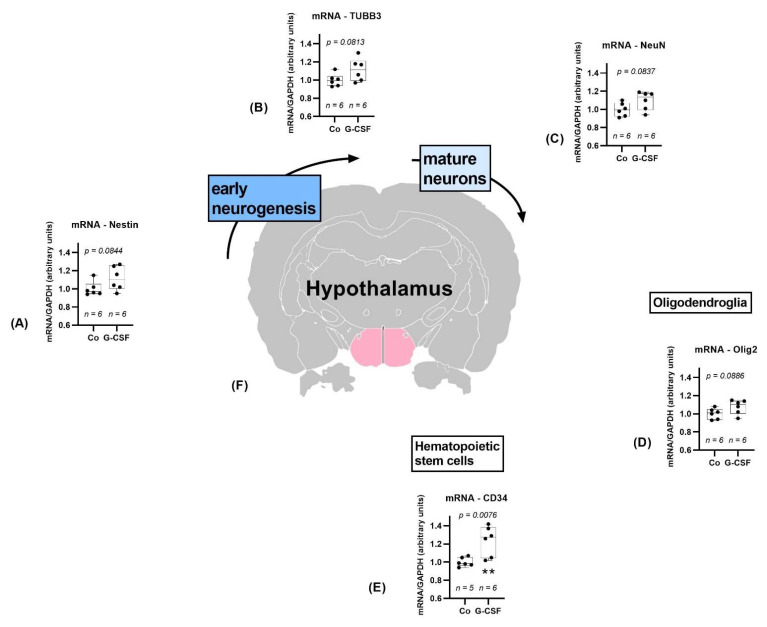
Study results of hypothalamus. Hypothalamus is marked pink in the cross section of the brain (**F**), position 2,07 mm posterior to bregma, image modified from Scalable Brain Atlas [36,37,38,39]. Controls are depicted left, G-CSF-treated animals are displayed on the right side of each graph. Neuronal differentiation markers Nestin (**A**), TUBB3 (**B**), and NeuN (**C**) show trends towards upregulation. Olig2 shows a trend towards upregulation (**D**). CD34 is significantly upregulated (**E**). All parameters were tested for Gaussian distribution using D’Agostino–Pearson omnibus normality test. Afterward, all parameters were analyzed using a two-tailed Student’s *t*-test or Mann–Whitney test, depending on Gaussian distribution. Data are presented as median with min to max. Significance was observed at *p* ≤ 0.01 (**), a trend was noticed at *p* ≤ 0.1. Numbers (n) are given for each group.

## Data Availability

Data is contained within the article and Appendix A.

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
