# Peer review of "Direct Potential Modulation of Neurogenic Differentiation Markers by Granulocyte-Colony Stimulating Factor (G-CSF) in the Rodent Brain"

_pharmaceutics, 2022, doi:10.3390/pharmaceutics14091858_

Round 1

Reviewer 1 Report

Comments to the Author

 The topic is of interest, and the manuscript is well illustrated. However, the clarity of the manuscript should be substantially improved.

I mean: (1) clarity in relation to the novelty of this article and (2) clarity of particular statements.

1. Are there controversies in this field? What are the most recent and important achievements in the field?  In my opinion, answers to these questions should be emphasized. Perhaps, in some cases, novelty of the recent achievements should be highlighted by indicating the year of publication in the text of the manuscript.

2. The results and discussion section is very weak and no emphasis is given on the discussion of the results like why certain effects are coming in to existence and what could be the possible reason behind them?

3. Results and conclusion. The section devoted to the explanation of the results suffers from the same problems revealed so far. Your storyline in the results section (and conclusion) is hard to follow. Moreover, the conclusions reached are really far from what one can infer from the empirical results. The discussion should be rather organized around arguments avoiding simply describing details without providing much meaning. A real discussion should also link the findings of the study to theory and/or literature.

4. Spacing, punctuation marks, grammar, and spelling errors should be reviewed thoroughly. I found so many typos throughout the manuscript.

 English is modest. Therefore, the authors need to improve their writing style. In addition, the whole manuscript needs to be checked by native English speakers.

Author Response

Dear reviewer,
Thank you for your comments. We think some of them are a bit hard to understand, but of course helpful - so we tried to improve the MS as much as we think is reasonable.

(1) "controversies": we touched the controversies in the translational aspects of its clinical relevance (more is not in the focus of the present pilot study).

(2) We did not want to over-interprete our data, but discussed our findings cautiously. We added some further possible explanations in the revised manuscript - in the discussion section - and clarifications. 

(3) In the results section we disaggree here, as we tried to line up the findings. The data presentation should be as transparent as possible. In the discussion we  triedd to be as cuatious as possible with the data derived from a pilot study. However, we chose to use a minimal number of experimental animals to get an interesting first insight and relevant safety information.

(4) Language and style. Thank you for your comments, we have worked on that accordingly. 

Best regards,
J.K. & U.B.

Reviewer 2 Report

The paper by Kozole et al. entitled “Direct Potential Modulation of Neurogenic Differentiation Markers by Granulocyte-Colony Stimulating Factor (G-CSF) in the Rodent Brain – a safety study” aims to address how a single supratherapeutic dose of G-CSF administered subcutaneously alters mRNA and protein expression of neurogenic and gliogenic markers in the rodent brain.

Animal brains were analyzed 5 days after a single injection of high-dose G-CSF at 1.3mg/kg or saline in the control group.

The authors report the expression of several neuronal and glial differentiation markers in various brain regions, including germinal zones (SVZ and hippocampus), white matter tracts, motor cortex, basal ganglia, cerebellum and hypothalamus.

The results show that stem cell and neuronal differentiation markers (e.g., Nestin, Sox2, MSI1, DCX, NCAM, tubulin and NeuN) were either downregulated or unaffected in most brain regions, including SVZ and hippocampus, but partially upregulated in cerebellum and hypothalamus. The same observations were made from the expression analyses of Olig2, TGFb, CD34, and CSF3R.

Overall, the study is well written and provides novel information to the field of neuroplasticity and how a common hematopoietic cytokine might influence neurogenesis and gliogenesis.

Comments:

·      The study is entitled “safety study” but it is not clear what “safety” aspects were evaluated. The claim that an expression analysis of several markers of glia and neuronal differentiation in brain regions after a single dose of G-CSF is a safety study is questionable and I would suggest to deleting “safety study” from the title. Moreover, multiple dose ranges and organ evaluation for toxicity are usually required for a toxicology and safety analysis - the current title is therefore misleading.

·      The authors make the case that G-CSF may be useful to promote neurogenesis and increase neural plasticity after neural injury. However, their findings support that neurogenesis and gliogenesis may in fact be downregulated after G-CSF. It therefore would be helpful to provide a discussion on this issue and the findings, and perhaps introduce a hypothesis in support of their findings. The authors may want to comment on the possibility whether the expression analysis of the markers of interest could have yielded different results if the analysis had been performed on other days, such as day 1, 3 or 10. Providing analysis from day 5 only appears to miss the point that protein and mRNA expression could be highly dynamic and perhaps initially upregulated and downregulated at later timepoints. 

·      It would be helpful to provide an explanation why the specific G-CSF dose of 1.3mg/kg was used and why the brain tissue analysis was done specifically after 5 days 

·      Did the authors find a correlation between leukocyte counts assessed daily in peripheral blood after G-CSF and their marker expression in the brain?

·      Fig 2: Ki-67 is considered a non-specific marker of cell proliferation and not specific to neuronal differentiation. The cartoon is misleading. Suggest to remove Ki67 from the table A and perhaps include in Table C ?

Author Response

Dear Reviewer, thank you for your comments and helpful remarks. 

(1) "safety": we omitted the term safety in the title, and discussed that issue subsequently. Concerning the enormous safety data existing for G-CSF in the clinics, we chose this study concept (short, pilot style, few animals) - so your remarks are absolutely well taken. 

(2) "downregulation of neuronal markers":again your remarks are absolutely well taken. We commented this issue mainly on lines 441 to 447, and 474-487. 

(3) "dose" / "5-day" timepoint: we intended to use an approx. 100 fold human dose, and clarified this issue on lines 490 - 496, where we also justified the 5-day timepoint of analysis. 

(4) Indeed a correlation of leukocyte counts with leukocyte markers in the observed brain regions would add interesting correlation data....unfortunately we looked for peripheral blood data over several days (to control the mobilization effect), but did not stain for leukocyte markers other than CD34, and that at day 5. 

(5) "Ki-67": absolutely right, we used that marker as a positive control marker, as it turns up as a general and neuronal marker in this context. We transplanted Ki-67 within Fig. 2 from A to C, as you suggested. 

In general we revised the manuscript according to your suggestions.
Cordially,
J.K. & U.B.

Round 2

Reviewer 1 Report

The author has addressed my all comments.